# Correlation between CD4 T-Cell Counts and Seroconversion among COVID-19 Vaccinated Patients with HIV: A Meta-Analysis

**DOI:** 10.3390/vaccines11040789

**Published:** 2023-04-04

**Authors:** Qian Zhou, Yihuang Liu, Furong Zeng, Yu Meng, Hong Liu, Guangtong Deng

**Affiliations:** 1Hunan Key Laboratory of Skin Cancer and Psoriasis, Department of Dermatology, Hunan Engineering Research Center of Skin Health and Disease, Xiangya Hospital, Central South University, Changsha 410008, China; 2National Clinical Research Center for Geriatric Disorders, Xiangya Hospital, Central South University, Changsha 410008, China; 3Department of Oncology, Xiangya Hospital, Central South University, Changsha 410008, China

**Keywords:** COVID-19, vaccines, HIV, CD4 T-cell

## Abstract

Objective: To evaluate the potential factors for predicting seroconversion due to the coronavirus disease 2019 (COVID-19) vaccine in people living with HIV (PLWH). Method: We searched the PubMed, Embase and Cochrane databases for eligible studies published from inception to 13th September 2022 on the predictors of serologic response to the COVID-19 vaccine among PLWH. This meta-analysis was registered with PROSPERO (CRD42022359603). Results: A total of 23 studies comprising 4428 PLWH were included in the meta-analysis. Pooled data demonstrated that seroconversion was about 4.6 times in patients with high CD4 T-cell counts (odds ratio (OR) = 4.64, 95% CI 2.63 to 8.19) compared with those with low CD4 T-cell counts. Seroconversion was about 17.5 times in patients receiving mRNA COVID-19 vaccines (OR = 17.48, 95% CI 6.16 to 49.55) compared with those receiving other types of COVID-19 vaccines. There were no differences in seroconversion among patients with different ages, gender, HIV viral load, comorbidities, days after complete vaccination, and mRNA type. Subgroup analyses further validated our findings about the predictive value of CD4 T-cell counts for seroconversion due to COVID-19 vaccines in PLWH (OR range, 2.30 to 9.59). Conclusions: The CD4 T-cell counts were associated with seroconversion in COVID-19 vaccinated PLWH. Precautions should be emphasized in these patients with low CD4 T-cell counts, even after a complete course of vaccination.

## 1. Introduction

The coronavirus disease 2019 (COVID-19) pandemic ravaged the globe. Vaccination has become the paramount method for the prevention of worse outcomes, including severe COVID-19 and death during or in SARS-CoV-2 infection [1]. People living with HIV(PLWH) are at higher risk of these worse outcomes due to immunosuppression and its comorbidities [2,3,4,5]. Therefore, PLWH are given priority for COVID-19 vaccination. Accumulating studies demonstrated that PLWH had a lower efficacy of COVID-19 vaccines than the general population [6,7,8]. With the implementation of supplemental vaccination among PLWH [9], it is important to identify the potential predictors for seroconversion due to the COVID-19 vaccine in PLWH. However, the potential factors for predicting seroconversion after the COVID-19 vaccine in PLWH were not completely understood.

Increasing individual studies have attempted to identify the factors associated with the effectiveness of vaccination against COVID-19 [10,11,12]. For example, Anais et al. [13] found that the seroconversion rate was slightly lower among PLWH with CD4 T-cell counts of <350 cell/mm^3^ and dramatically reduced among those with CD4 T-cell counts of <200 cell/mm^3^. Moreover, the type of vaccine, the presence of comorbidities, and HIV status might preclude the development of a protective immunological response. However, the lack of studies with large sample sizes highlights the importance of a comprehensive analysis of the potential predictors for seroconversion due to the COVID-19 vaccine in PLWH.

Thus, this systematic review and meta-analysis was to provide a review of the potential factors associated with seroconversion to COVID-19 vaccines among PLWH. We extracted data from observational studies, and the factors included age, gender, CD4 T-cell count, HIV viral load, comorbidities, days after complete vaccination, and vaccine type and were evaluated with respect to the seroconversion rate in PLWH after COVID-19 vaccination. Our findings will help to plan better prevention strategies for this frail population.

## 2. Methods

We conducted a systematic review and meta-analysis following the Preferred Reporting Items for Systematic Reviews and Meta-analyses (PRIMSA) guidelines [14]. The protocol of our meta-analysis has been submitted to the International Prospective Register of Systematic Reviews (PROSPERO). The registration number of PROSPERO is CRD42022359603.

### 2.1. Search Strategy

The PubMed, Embase and Cochrane Library databases were searched for relevant studies from the databases’ inception to 13th September 2022 using the following search terms: (corona[ti] OR covid*[ti] OR sars[ti] OR severe acute respiratory syndrome[ti] OR ncov*[ti] OR “severe acute respiratory syndrome coronavirus 2” [Supplementary Concept] OR “COVID-19” [Supplementary Concept] OR (wuhan[tiab] AND coronavirus[tiab]) OR (wuhan[tiab] AND pneumonia virus[tiab]) OR COVID19[tiab] OR COVID-19[tiab] OR coronavirus 2019[tiab] OR SARS-CoV-2[tiab] OR SARS2[tiab] OR SARS-2[tiab] OR “severe acute respiratory syndrome 2”[tiab] OR 2019-nCoV[tiab] OR (novel coronavirus[tiab] AND 2019[tiab]) NOT (animals[mesh] NOT humans[mesh])) AND (“Vaccines”[MeSH] OR “vaccination”[MeSH] OR vaccine[All Fields] OR vaccination[All Fields] OR vaccin*[All Fields]) AND (“HIV Infections” [MeSH] OR “HIV”[MeSH] OR “hiv”[tw] OR hiv infect*[tw] OR “human immunodeficiency virus”[tw] OR “human immunedeficiency virus”[tw] OR “human immuno-deficiency virus”[tw] OR “human immune-deficiency virus”[tw] OR ((human immun*) AND (“deficiency virus”[tw])) OR “acquired immunodeficiency syndrome”[tw] OR “acquired immunedeficiency syndrome”[tw] OR “acquired immuno-deficiency syndrome”[tw] OR “acquired immune-deficiency syndrome”[tw] OR ((acquired immun*) AND (“deficiency syndrome”[tw]))). No language restrictions were imposed. The full details of the search strategies can be found in Appendix A.

### 2.2. Inclusion and Exclusion Criteria

The study selection was conducted in three steps: removing the initial deduplication, screening titles and abstracts, and reviewing the full text of the potentially eligible articles. Two reviewers (Q.Z. and Y.L.) independently evaluated eligibility, and the discrepancies were solved by a third investigator (G.D.). Articles were included for analysis if they met the following criteria: (1) cohort studies or randomized controlled trials; (2) patients living with HIV; (3) the odds ratio (OR) of potential predictors of seroconversion due to COVID-19 vaccine was reported, or the OR could be calculated according to the data from the studies. There were no restrictions regarding age, sex or duration of the study. The cohort studies were defined as those that sampled participants based on exposure, followed-up participants over time, and ascertained the outcomes [15]. Case reports, case series, and studies with data inaccessible from the corresponding authors were excluded.

### 2.3. Data Extraction and Quality Assessment

Two investigators (Q.Z. and Y.L.) independently extracted data based on a predetermined proforma in Microsoft Excel. The following information was recorded for the studies: the first author, publication year, country, study type, data source, patient number, vaccine type, vaccine dose, potential predictors for seroconversion due to the COVID-19 vaccine, multivariable analysis, COVID-19 history, and adjustment parameters. We used the Cochrane Risk of Bias 2 tool to assess the risk of bias for the randomized controlled studies [16], while the Risk of Bias in Nonrandomized Studies of Interventions (ROBINS-I) tool was used for the comparative cohort studies [17]. For the Cochrane Risk of Bias 2 tool, the risk of bias judgment per study is noted as low risk when all the domains are judged as being a low risk of bias, noted with some concerns when one or more domains are judged as some concerns, or high risk when at least one domain is judged as being at a high risk of bias, or when multiple domains are judged as some concerns. During our search, 17 eligible comparative cohort studies were included. For the ROBINS-I tool, the risk of bias judgment per study is noted as low risk when all domains are judged as being at low risk of bias, moderate risk when one domain is judged as a moderate risk of bias, serious risk when one domain is judged as serious risk of bias, or critical risk of bias when one domain is judged as being at critical risk of bias. Only one randomized study was founded in our study. The risk of bias for non-comparative cohort studies was regarded as a high risk of bias. We rated the quality of evidence according to the grading of recommendations, assessment, development and evaluation (GRADE) approach to assess the certainty of the evidence obtained from the present meta-analysis of potential risk factors of seroconversion among COVID-19 vaccinated PLWH.

### 2.4. Definitions of Vaccines

The inactivated vaccines included BBIBP-CorV, Corona Vac or Sinopharm; the mRNA vaccines comprised the BNT162b2 or mRNA-1273; the adenovirus vaccines comprised the ChA-dOx1 nCoV-19 or Ad.26.COV2.S; and mixed vaccines mean more than one type of vaccine.

### 2.5. Statistical Analysis

The primary outcome was the odds ratio and its corresponding 95% CI of potential predictors for seroconversion to the COVID-19 vaccine in PLWH. If the outcomes were presented as RRs, data were converted to the ORs for analysis by using the formula OR = RR(1-pRef)/(1-RR×pRef), where pRef is the prevalence of the outcome in the control [18]. The *p*-value by χ^2^ test < 0.1 or the I^2^ statistic was ≥ 50% and was considered to indicate significant heterogeneity among the included studies. In this case, the pooled odds ratios were estimated by the fixed-effects model; otherwise, the random-effects model was preferentially performed. Subgroup analyses were conducted to evaluate the predictive value of the CD4 T-cell counts for seroconversion due to the COVID-19 vaccine according to the study location (Europe vs. America vs. Asia), study design (retrospective vs. prospective), source of data (multi-center vs. single-center), sample size (< 100 vs. ≥ 100), cut off of CD4 T-cell counts (200 cell/mm^3^ vs. 500 cell/mm^3^ vs. others), vaccine type (inactivated vaccine vs. mRNA vaccine vs. mix), and multivariable analysis (YES vs. NO). Meta-regression analyses were further performed to explore the potential effect of these parameters on the outcomes. The regression coefficient was calculated to describe the change in outcomes with explanatory variables (potential effect modifiers). The potential publication bias was evaluated by Egger’s test, and funnel plots were drawn if the studies were above 10. Trim-and-fill analyses were performed to adjust for publication bias (Egger’s test *p* < 0.05). Sensitivity analyses were conducted where the outcomes were recalculated by omitting one study at a time. All calculations and graphs were performed and visualized with R statistic software (3.6.3).

## 3. Results

### 3.1. Study Selection, Characteristics and Quality Assessment

The study selection is shown in Figure 1. A total of 1592 potentially relevant studies were identified through the literature search. After screening the initial titles and abstracts, the full text of 63 studies was further considered for eligibility. After the removal of another 40 studies, including 33 studies that failed to report the odds ratios, 5 cross-sectional studies, and 2 reviews (Appendix A), 23 studies [13,19,20,21,22,23,24,25,26,27,28,29,30,31,32,33,34,35,36,37,38,39,40] that included 4428 patients living with HIV were finally included in the meta-analysis (Figure 1). 

The main characteristics and clinical outcomes of the studies for quantitative analysis are summarized in Table 1. Of these studies, 11 were from Europe, 5 were from Asia, 5 were from North America, 1 was from South Africa, and 1 was from South America. The studies comprised 14 prospective studies and 9 retrospective studies. A total of 7 studies were multi-center, and 16 were single-center. The number of PLWH in 14 studies was above 100; 6 studies had adjusted for potential confounders; 7 studies were analyzed using multivariable analysis; all PLWH in the 19 studies were not infected with COVID-19 prior to vaccination. In terms of vaccination type, the mRNA vaccines were used in 11 studies; adenovirus vaccines were used in 1 study; inactivated vaccines were used in 6 studies; and another 5 studies involved two or more vaccines or other types of vaccines. PLWH, in 2 studies, received an incomplete dose of vaccines; 20 studies received a complete dose of vaccines; only 1 study received the booster dose of vaccines. Appendix A shows the detailed risk of bias for each study, and most of the studies were regarded as critical or at a high risk of bias. 

### 3.2. Risk Factors for Seroconversion Rate in PLWH

A summary of the findings from the included studies is shown in Figure 2. A total of 7, 7, 16, 7, 3, and 7 studies provided data for the age, gender, CD4 T-cell counts, HIV viral load, comorbidities and days after complete vaccination, respectively (Appendix A). The pooled data showed that there were no statistical differences in seroconversion among patients with different ages, gender, HIV viral load, comorbidities and days after their complete vaccination. Notably, seroconversion was about 4.6 times in patients with higher CD4 T-cell counts (OR = 4.64, 95% CI 2.63 to 8.19) compared with those with lower CD4 T-cell counts. Quantitative synthesis was also accessible for different vaccine types, including mRNA vaccines vs. other vaccines and mRNA-1273 vs. BNT126b2 vaccines (another mRNA vaccine) (Appendix A). The pooled data demonstrated that there was no difference in seroconversion between patients receiving the mRNA-1273 vaccines and those receiving the BNT126b2 vaccines. It is worth noting that seroconversion was about 17.5 times in patients receiving mRNA COVID-19 vaccines (OR = 17.48, 95% CI 6.16 to 49.55), compared with those receiving other types of COVID-19 vaccines. 

### 3.3. Publication Bias

Egger’s test detected the existence of publication bias in potential predictors for seroconversion due to COVID-19 vaccines, including the CD4 T-cell counts (*p* < 0.01) and HIV viral load (*p* = 0.02). The funnel plot also showed the relative asymmetry in CD4 T-cell counts (Appendix A). After eight studies were filled, the funnel plot showed relative symmetry (Appendix A), and Egger’s test showed no evidence of significant publication bias (*p* = 0.60). Patients with high CD4 T-cell counts still had higher seroconversions than those with low CD4 T-cell counts (OR = 1.85, 95% CI 1.05 to 3.28). As for the HIV viral load, after three studies were filled, Egger’s test showed no evidence of significant publication bias (*p* = 0.78) (Appendix A) with still no statistical difference in the seroconversions among patients with different HIV viral loads (OR = 1.30, 95% CI 0.40 to 4.21).

### 3.4. Meta-Regression and Subgroup Analysis

Considering the potential predictive value of the CD4 T-cell counts in seroconversion due to the COVID-19 vaccines in PLWH, univariate meta-regression and subgroup analyses were further carried out to explore the source of heterogeneity. The univariate meta-regression found no significant moderators of heterogeneity (Appendix A). All the subgroup analyses arrived at a consistent conclusion (OR range, 2.30 to 9.59) (Figure 3 and Appendix A). Interestingly, a subgroup analysis, conducted according to the cutoff of CD4 T-cell counts, demonstrated that the odds ratio was highest in the cutoff for 200 cell/mm^3^ (OR = 6.18, 95% CI 2.98 to 12.84), followed by the cutoff for others (OR = 5.80, 95% CI 2.04 to 16.48), and a cutoff for 500 cell/mm^3^ (OR = 2.30, 95% CI 1.45 to 3.64) (*p* = 0.04) (Figure 3 and Appendix A). Subgroup analysis, stratified by vaccine type, showed that the odds ratio was lowest when patients received the inactivated vaccine (OR = 2.90, 95% CI 1.64 to 5.11), followed by the mRNA vaccine (OR = 5.38, 95% CI 1.77 to 16.32), and mixed vaccines-type (OR = 8.76, 95% CI 4.81 to 15.95) (*p* = 0.03) (Figure 3 and Appendix A). No significant heterogeneity was observed in the other subgroup comparisons (all *p* > 0.05).

### 3.5. Sensitivity Analysis

The sensitivity analysis, performed by using the “leave-one-out”, did not markedly change our results in these comparisons (Appendix A).

### 3.6. Grading the Quality of Evidence

According to the GRADE approach, the quality of evidence was very low for age, gender, CD4 T-cell counts, HIV viral load, comorbidities, days after complete vaccination, and vaccine-type impact factors (Appendix A). 

## 4. Discussion

In this meta-analysis, 23 studies with a total of 4428 patients living with HIV were included. We evaluated the potential factors for predicting seroconversion due to the COVID-19 vaccine in PLWH. We demonstrated that the CD4 T-cell counts and mRNA vaccines are associated with seroconversion due to COVID-19 vaccination. Compared with the PLWH with lower CD4 T-cell counts, the seroconversion rate was about 4.6 times in patients with higher CD4 T-cell counts. The mRNA COVID-19-vaccinated PLWH showed about 17.5 times the seroconversion compared with those receiving other types of COVID-19 vaccines, such as the inactivated vaccines. Advanced age, gender, HIV viral load, comorbidities, days after complete vaccination, and different mRNA vaccine types showed no association with seroconversion. The subgroup analysis further validated our finding about the predictive value of CD4 T-cell counts for seroconversion in PLWH. 

An effective COVID-19 vaccination strategy becomes the main measure of reducing the risk and mortality of COVID-19 [1]. Immunocompromised patients are of particular interest because of their attenuated responses to various vaccines [41,42,43]. There are several meta-analyses reporting seroconversion due to the COVID-19 vaccine in immunocompromised patients, including PLWH. For example, Mehrabi Nejad et al., demonstrated that immunocompromised patients had a lower overall crude prevalence of seroconversion. Notably, transplant patients were less likely to develop seroconversion after both the first and second dose compared with patients with malignancy or autoimmune disease [44]. However, this study did not include PLWH and also failed to evaluate the risk factors for seroconversion. Yin et al., mainly focused on the PLWH and reported that a second dose could consistently improve the seroconversion, although the seroconversion is still lower in PLWH than in healthy individuals [45]. Conversely, Kang et al., considered the immunogenicity and safety of the COVID-19 vaccine in PLWH to be acceptable because there were no significant differences in the seroconversion rates and incidence rates of adverse events of COVID-19 vaccines between PLWH and healthy controls [46]. However, these two studies still failed to evaluate the risk factors for seroconversion in PLWH. In our study, we comprehensively evaluated the potential risk factors for seroconversion due to the COVID-19 vaccine in PLWH, including age, gender, CD4 T-cell counts, HIV viral load, comorbidities, days after complete days, and vaccine type. We finally found that the CD4 T-cell counts and vaccine type are associated with seroconversion due to COVID-19 vaccination. Our findings were completely different from previous meta-analyses and filled a gap in the risk factors for seroconversion due to the COVID-19 vaccine. PLWH are characterized by impaired immunity with reduced CD4 T-cell counts. For the generation of antibody response to vaccination, an essential step is the interaction of antigen-primed B cells and CD4 T-cells in the germinal center reaction, where CD4 T-cells provide critical helper function for the B cells to undergo proliferation, isotype switching and somatic hypermutation [47,48]. Despite antiretroviral therapy, the immune dysfunction may not be completely reversed [13]. Therefore, PLWH could have decreased response to vaccination due to defects of CD4 T-cells’ help [49,50]. Previous studies showed considerably weaker responses among PLWH with CD4 T-cell counts of < 300 cells/mm^3^ compared with HIV-negative individuals [19,26,30,51]. Some studies found similar humoral immune responses in PLWH with CD4 T-cell counts of > 500 cells/mm^3^ compared to the health controls [19,31]. Here, we found that the CD4 T-cell counts are associated with seroconversion in COVID-19-vaccinated PLWH. More importantly, we found that the odds ratio was higher in the cutoff for 200 cell/mm^3^ than in the cutoff for 500 cell/mm^3^. These results suggested that lower CD4 T-cell counts predict lower seroconversion in COVID-19-vaccinated PLWH. 

It is also worth noting that the mRNA vaccine developed the highest serologic response in PLWH than the inactivated and adenovirus vaccines, which suggested PLWH are given priority for the mRNA vaccine. This finding was consistent with the results from general patients [52,53]. For example, previous studies demonstrated that compared with the mRNA vaccine, the antibody level of inactivated CoronaVac-vaccinees wanes quickly, and patients after the vaccine face a higher risk of breakthrough infection [54,55]. Further, the geometric NAbTs of inactivated vaccinees were 19-fold lower than that of the BNT162b2-vaccines [56,57]. The mRNA vaccines also showed superior cellular immune responses when compared to other vaccines against SARS-CoV-2 [58]. The median levels of CD4 responses following the mRNA vaccine were higher than the adenoviral vaccine, followed by the inactivated vaccine [58]. In a recent UK population-based study, the mRNA BNT162b2 vaccine showed to be more efficacious than the adenovirus ChA-dOx1 nCoV-19 vaccine against SARS-CoV-2 infection and hospital admissions for COVID-19 [59]. Our findings suggest that, just like the general population, PLWH respond better to the mRNA vaccines than other vaccines.

This systematic review and meta-analysis had some limitations. First, some of the studies included were observational studies, which might cause a risk of unbalanced groups for comparison with a high risk of bias. Second, significant heterogeneity and publication bias were found in some analyses, while the outcomes of trim-and-fill analyses, sensitivity analyses, and subgroup analyses were consistent. Third, ART is fundamental to the clinical care of PLWH, while the association between ART and seroconversion in PLWH was not evaluated due to a lack of data. Fourth, we did not evaluate the predictive value of the nadir CD4 counts for the seroconversion in PLWH due to data unavailability. Finally, the subgroup analysis was only performed for the CD4 T-cell counts but not other potential predictors due to fewer than 10 studies. Numerous studies are needed to pool the potential predictors for seroconversion due to the COVID-19 vaccine in PLWH.

## 5. Conclusions

CD4 T-cell counts are associated with seroconversion in COVID-19-vaccinated PLWH. Precautions should be emphasized in these patients with low CD4 T-cell counts, even after a complete vaccination course. Moreover, the mRNA vaccines might be a priority for PLWH with COVID-19.

## Figures and Tables

**Figure 1 vaccines-11-00789-f001:**
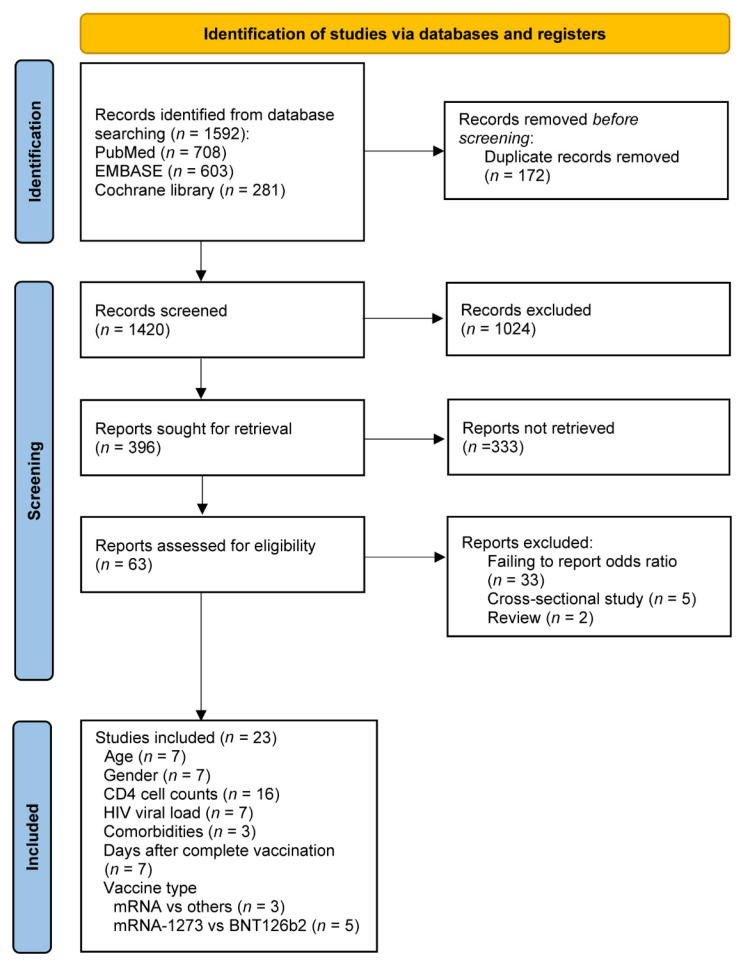
Flowcharts illustrating the article selection process.

**Figure 2 vaccines-11-00789-f002:**
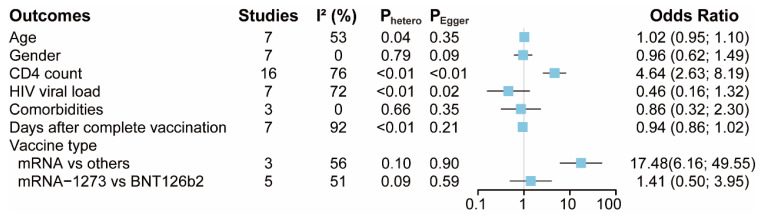
The pooled odds ratio of potential factors associated with seroconversion rate in people living with HIV(PLWH). Comorbidities mean PLWH with cirrhosis, HBV or HCV coinfections.

**Figure 3 vaccines-11-00789-f003:**
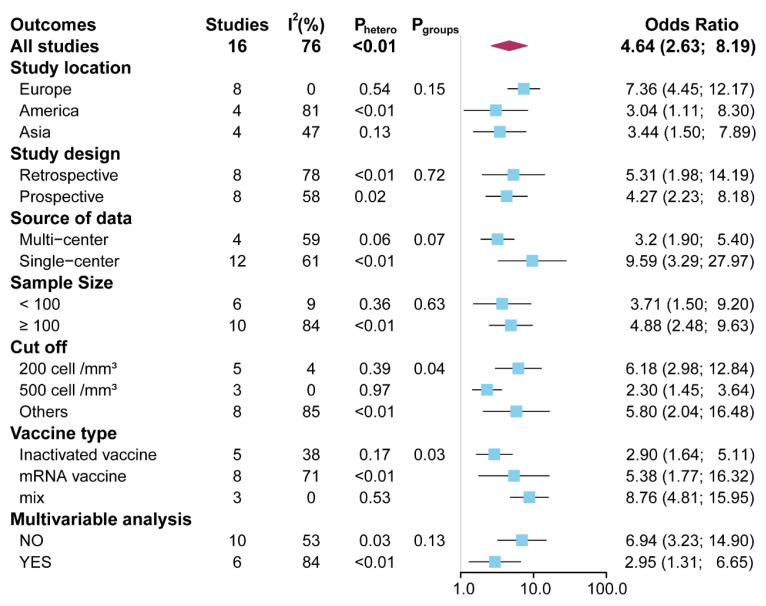
The pooled odds ratio of CD4 T-cell counts in subgroup analyses.

**Table 1 vaccines-11-00789-t001:** Characteristics of included studies.

Source	Country	Design	Data Source	Cases	Vaccine Type	Vaccine Dose	COVID-19 History	Outcomes	Multivariable Analysis	Impact Factors	Adjust
Anais 2022 [13]	Spain	Pro	Three university hospitals in Southern Spain	420	BNT162b2, mRNA-1273, ChAdOx1 nCoV-19, or Ad26.COV2.S	Complete	NO	anti-S IgG	YES	Age, gender, HIV infection way, CDC clinical category nadir CD4 T-cell counts, Charlson index, cirrhosis, chronic kidney disease, immunosuppressive therapy, CD4 T-cell counts (cutoff = 200 cell/mm^3^), HIV viral load, vaccine	-
Antinori 2022 [19]	Italy	Pro	National Institute for Infectious Diseases Lazzaro Spallanzani	153	BNT162b2 or mRNA-1273	Complete	NO	nAbs	NO	CD4 T-cell counts (cutoff = 200 cell/mm^3^)	-
Ao 2022 [20]	China	Pro	People’s Hospital of Tongliang District	139	BBIBP-CorV or Corona Vac	Complete	NO	anti-RBD IgG	YES	Age, gender, days after 2nd vaccination, CD4 T-cell counts (cutoff = 500 cell/mm^3^), HIV viral load, white blood cell count, lymphocyte count, platelet count, alanine aminotransferase, aspartate aminotransferase, B cells, RBD-specific B cells, RBD-specific MBCs, RBD^+^ rMBCs, RBD^+^ actMBCs, RBD^+^ atyMBCs, RBD^+^ intMBCs	-
Bergman 2021 [21]	Sweden	Pro	Karolinska University Hospital	79	BNT162b2	Complete	NO	anti-RBD IgG	NO	CD4 T-cell counts (cutoff = 300 cell/mm^3^)	Age (partially)
Brumme 2022 [22]	Canada	Retro	Three HIV care clinics in	100	BNT162b2, mRNA-1273 or ChAdOx1	Complete	YES	anti-RBD IgG	NO	Days after 2nd vaccination	Age, chronic health conditions
Gianserra 2022 [23]	Italy	Pro	HIV/AIDS Unit of the San Gallicano Dermatological Institute	42	BNT162b2	Complete	NO	SARS-CoV-2 S1/S2 IgG	NO	Days after second vaccination	-
Haidar 2022 [24]	USA	Pro	Unive University of Pittsburgh Medical Center Health System	94	BNT162b2, mRNA-1273 or Adenovirus	Complete	NO	anti-RBD IgG	NO, except for days after 2nd dose	Age, gender, race, vaccine, days after second dose	-
Han 2022 [25]	China	Retro	Beijing Ditan Hospital	47	CoronaVac or Sinopharm	Complete	NO	nAbs	NO	CD4 T-cell counts (cutoff = 350 cell/mm^3^)	Age, sex, and interval length
Hassold 2022 [26]	France	Retro	Department of Infectious Diseases of Hospital Avicenne	105	BNT162b2, mRNA-1273 or ChAdOx1-nCoV-19	Complete	NO	Anti-spike IgG	NO	CD4 T-cell counts (cutoff = 200 cell/mm^3^)	-
Hensley 2022 [27]	Netherlands	Pro	22 HIV treatment centers	1154	BNT162b2, mRNA-1273, ChAdOx1-S or Ad26.COV2.S	Complete	NO	Anti-spike IgG	YES, except for vaccine type	Vaccine type, age, gender, HIV viral load, CD4 T-cell counts (cutoff = 250 cell/mm^3^), CD4 nadir cell counts	-
Khan 2022 [28]	South African	Pro	Biomedical Research of the University of KwaZulu–Natal	26	Ad26.CoV2.S	Complete	YES	Neutralization capacity	NO	HIV viral load	-
Milano 2022 [29]	Italy	Pro	University of Bari	578	BNT162b2	Complete	NO	Anti-RBD IgG	NO	Days after complete vaccination	-
Nault 2022 [30]	Canada	Retro	HIV clinics in Montreal	106	mRNA-1273	Uncomplete	YES	Anti-RBD IgG	NO	CD4 T-cell counts (cutoff = 250 cell/mm^3^)	-
Netto 2022 [31]	Brazil	Pro	University of Sao Paulo HIV/AIDS outpatient clinic	215	CoronaVac	Complete	NO	nAbs	NO	CD4 T-cell counts (cutoff = 500 cell/mm^3^)	-
Polvere 2022 [32]	Italy	Retro	Azienda Ospedaliera Universitaria Senese	84	BNT162b2 or mRNA-1273	Complete	NO	nAbs	NO	Age, gender, vaccine type, BMI, IDU, years from HIV infection, CDC stage, HBV or HCV coinfection, zenith HIV-RNA, CD4 T-cell counts at nadir, years from first ART, type of ART, HIV viral load, time from last HIV-RNA >50 copies/mL, CD4 T-cell counts at baseline (cutoff = 350 cell/mm^3^), CD4%, CD4/CD8 ratio	-
Speich 2022 [33]	Switzerland	RCT	University Hospital Basel, University Hospital Bern and University Hospital Zurich	341	BNT162b2 or mRNA-1273	Complete	YES	nAbs	NO	Vaccine type	RCT
Spinelli 2022 [34]	USA	Retro	A large outpatient HIV clinic	100	BNT162b2 or mRNA-1273	Complete	NO	nAbs	YES	CD4 T-cell counts (cutoff = NA), HIV viral load, vaccine type	Care for chronic medical conditions on days since completion of second vaccination (minimum 10), sex, age and mRNA vaccine type
Tuan 2022 [35]	USA	Retro	Two HIV clinics of the Yale New Haven Health System	78	BNT162b2	Uncomplete	NO	IgG	NO, except for CD4 T-cell counts	Age, gender, days after second vaccination, BMI, self-reported substance use, time since HIV diagnosis, HIV ART regimen, CD4 T-cell counts (cutoff = 500 cell/mm^3^), HIV viral load, comorbidities	-
Vergori 2022 [36]	Italy	Retro	Infectious Diseases Lazzaro Spallanzani in Rome	106	BNT162b2 or Mrna-1273	Booster	NO	nAbs	NO	CD4 T-cell counts (cutoff = 200 cell/mm^3^), CD4 T-cell counts at nadir	-
Wong 2022 [37]	China	Pro	The Integrated Treatment Centre or Princess Margaret Hospital HIV Service	213	CoronaVac or Comirnaty	Complete	NO	nAbs	NO	Vaccine type	age, sex, CD4 T-cell counts, and suppressed viral load (SVL) at the time point nearest to vaccination.
Xu 2022 [38]	Sweden	Pro	Karolinska University Hospital	79	BNT162b2	Complete	NO	anti-spike-IgG	NO	CD4 T-cell counts (cutoff = 200 cell/mm^3^)	-
Zeng 2022 [39]	China	Retro	The Third People’s Hospital of Shenzhen	126	BBIBP-CorV or CoronaVac	Complete	NO	anti-RBD IgG	NO	CD4 T-cell counts (cutoff = 350 cell/mm^3^), days after complete vaccination, vaccine type	-
Zou 2022 [40]	China	Pro	Wuchang district of Wuhan city	46	Sinopharm WIBP-CorV	Complete	NO	nAbs and IgG	YES, except for days after 2nd dose	Age, gender, CD4 T-cell counts (cutoff = NA), days after second dose	-

**Abbreviations:** COVID-19, coronavirus disease 2019; Pro, prospective study; Retro, retrospective study; RCT, randomized controlled trial; RBD, receptor binding domain; Ig, immunoglobulin; S, spike; nAbs, neutralizing antibodies; BMI, body mass index; MBC, memory B cell; BV, hepatitis B virus; HCV, hepatitis C virus; IDU, injecting drug users; NA, not available.

## Data Availability

The data supporting this study’s findings are available from the corresponding author upon reasonable request.

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
