# Peer review of "Correlation between CD4 T-Cell Counts and Seroconversion among COVID-19 Vaccinated Patients with HIV: A Meta-Analysis"

_vaccines, 2023, doi:10.3390/vaccines11040789_

Round 1

Reviewer 1 Report

1- Difference between various type of vaccines against COVID was not discussed, please add it.

2- CD4 counts can be associated with sero by covid-19 patients?

Author Response

-Difference between various type of vaccines against COVID was not discussed, please add it.

Response: Thanks for your excellent suggestion. The related discussion has been added in the discussed section.

“mRNA vaccines also showed the superior cellular immune responses compared to other vaccines against SARS-CoV-2[52]. Median levels of CD4 responses following mRNA vaccine were higher than adenoviral vaccine, followed by inactivated vaccine [52]. In a recent UK population-based study, the mRNA BNT162b2 vaccine showed more efficacious than the adenovirus ChA-dOx1 nCoV-19 vaccine against infection of SARS-CoV-2 and hospital admissions for COVID-19 [53].”

Reference:

  1. Ben Ahmed M, Bellali H, Gdoura M, et al. Humoral and Cellular Immunogenicity of Six Different Vaccines against SARS-CoV-2 in Adults: A Comparative Study in Tunisia (North Africa). Vaccines (Basel) 2022;10(8) doi: 10.3390/vaccines10081189 [published Online First: 2022/07/28]
  2. Wei J, Zhang W, Doherty M, et al. Comparative effectiveness of BNT162b2 and ChAdOx1 nCoV-19 vaccines against COVID-19. BMC Med 2023;21(1):78. doi: 10.1186/s12916-023-02795-w [published Online First: 2023/03/02]

-CD4 counts can be associated with sero by covid-19 patients?

Response: Thanks for your questions. After scanning numerous COVID-19 papers, we did not find the association between CD4 counts and seroconversion in general COVID-19 patients. However, there is a strong association in COVID-19 PLWH (Netto LC, Ibrahim KY, Picone CM, et al. Safety and immunogenicity of CoronaVac in people living with HIV: a prospective cohort study. Lancet HIV. 2022;9(5):e323-e331. doi:10.1016/S2352-3018(22)00033-9). This report was also consistent with our findings in our manuscript. Our manuscript is the first meta-analysis to evaluate the association between CD4 counts and seroconversion in COVID-19 PLWH. Therefore, our manuscript is of great significance. Thanks a lot.

Reviewer 2 Report

Review

Correlation between CD4 Cell Counts and Seroconversion among COVID-19 Vaccinated Patients with HIV: A

Meta-Analysis, by Qian Zhou et al.

Section1 : Introduction

General observation: People living with HIV should be PLWH.

CD4 T cells instead of CD4 cells.

Line 33: after outcomes should be added “during or in SARS-CoV-2 infection,”

Line 33: thought to be should be cancelled. PLWH are at a high risk to develope a severe COVID.

Section 2: Methods

Line 61: SARS-CoV-2 instead of SARS-Cov-2

Section Discussion:

line 194: CD4 T cells count instead of CD4 cells.

Author Response

Section1: Introduction

-General observation: People living with HIV should be PLWH.

Response: Thanks for your suggestions. We have changed the phrase “People living with HIV” to the general observation PLWH in the revised manuscript. Thanks a lot.

-CD4 T cells instead of CD4 cells.

Response: Thank you for pointing out this issue. we have corrected it in the revised manuscript. Thanks a lot.

-Line 33: after outcomes should be added “during or in SARS-CoV-2 infection,”

Response: Thanks for your suggestions. As you suggested, we have added it in our revised manuscript.

-Line 33: thought to be should be cancelled. PLWH are at a high risk to develop a severe COVID.

Response: Thanks for your advice. We have cancelled them in our revised manuscript. Thanks a lot.

Section 2: Methods

-Line 61: SARS-CoV-2 instead of SARS-Cov-2

Response: Thanks for your carefulness. We have made corrections as you suggested. Thanks a lot.

Section Discussion:

-line 194: CD4 T cells count instead of CD4 cells.

Response: Thank you for pointing it out, and we have corrected it in the revised manuscript. Thanks a lot.

Reviewer 3 Report

The manuscript is well written but impossible for anyone not experienced with the software to follow.

Conceptual problems

·               It is not clear whether all patients were on ART and had been on for sufficient time to experience CD4 recovery and achieve a low VL. This may explain the lack of significant associations with VL. ART is fundamental to clinical care.

·               On finding an association with CD4 counts, it would be useful and interesting to look at nadir CD4 counts. This could guide physicians treating patients with low nadir count

·               How are high and low CD4 counts defined?

·               What “comorbidities” were included and what were the criteria for diagnosis?

·               Were inactivated vaccines mostly given to patients with lower CD4 counts &/or more comorbidities (ie: in the developing world)?

·               Are nAb assays as sensitive as IgG ELISAs?

Presentation issues

·               There are a lot of Supplementary figures and most readers would deduce nothing from them as they lack informative (explanatory) legends. Are they all needed?

·               What sort of vaccines are BNT162b2 and BBIBP-CorV. This isn’t provided. You also do not define “mix”

·               The initial inclusion of “Wagner A, Garner-Spitzer E, Schötta AM, et al” was clearly your error. It should be omitted.

·               Table S4 lacks an informative legend

Author Response

-The manuscript is well written but impossible for anyone not experienced with the software to follow.

Response: Thanks for your appreciation for our writing. R software is used for meta-analysis in many excellent papersBentzley, B. S., Han, S. S., Neuner, S., Humphreys, K., Kampman, K. M., & Halpern, C. H. (2021). Comparison of Treatments for Cocaine Use Disorder Among Adults: A Systematic Review and Meta-analysis. JAMA network open, 4(5), e218049.. The results from R software could also be replicated using RevMan or STATA software. Therefore, we do not consider the use of software as a defect in our manuscript. Thanks a lot.

Conceptual problems

-It is not clear whether all patients were on ART and had been on for sufficient time to experience CD4 recovery and achieve a low VL. This may explain the lack of significant associations with VL. ART is fundamental to clinical care.

Response: Thanks for your comments. As you suggested, ART is fundamental to clinical care. However, there is a lack of studies exploring the difference between PLWH with and without ART therapy. Besides, the papers we included in our meta-analysis did not report whether patients had enough time to experience CD4 T cell recovery and achieve a low VL. Therefore, we have added the comments in our limitations: " Third, ART is fundamental to clinical care in PLWH, while the association between ART and seroconversion in PLWH was not evaluated due to lack of data." Thanks a lot for your valuable comments.

-On finding an association with CD4 counts, it would be useful and interesting to look at nadir CD4 counts. This could guide physicians treating patients with low nadir count.

Response: Thanks for your valuable suggestions. We also think it useful and interesting to look at nadir CD4 counts for the predictive value for the seroconversion in PLWH. However, only two papers reported the outcomes. Thus, we could not evaluate them. We have added the comments in our limitations: “Fourth, we did not evaluate the predictive value of nadir CD4 counts for the seroconversion in PLWH due to data unavailability.” Thanks a lot for your valuable comments.

-How are high and low CD4 counts defined?

Response: Thanks for your comments. The definition of high and low CD4 T cell counts is based on the cutoff in individual study. We have added the cutoff of CD4 T cell in the revised Table 1.

- What “comorbidities” were included and what were the criteria for diagnosis?

Response: Thanks for your questions. Three studies reported the seroconversion between PLWH with and without comorbidities which included cirrhosis, HBV or HCV coinfection. Thus, PLWH with comorbidities means those with cirrhosis, HBV or HCV coinfection in our manuscript. We have added the description in our revised Figure 2 legend to make our data more readable. Thanks a lot.

-Were inactivated vaccines mostly given to patients with lower CD4 counts &/or more comorbidities (ie: in the developing world)?

Response: Thank you for questions. There is no association between the use of inactivated vaccines and CD4 counts or comorbidities. Notably, inactivated vaccines were all used in developing countries including China and Brazil. We have added the description in our revised results. Thanks a lot.

-Are nAb assays as sensitive as IgG ELISAs?

Response: Thanks for the questions. To answer your questions, we checked tons of papers. Luckily, we found the evidence that IgG ELISAs is more sensitive than nAb assays. You could check the details in the paper (Dolscheid-Pommerich R, Bartok E, Renn M, et al. Correlation between a quantitative anti-SARS-CoV-2 IgG ELISA and neutralization activity. J Med Virol. 2022;94(1):388-392. doi:10.1002/jmv.27287). Thanks a lot.

Presentation issues

-There are a lot of Supplementary figures and most readers would deduce nothing from them as they lack informative (explanatory) legends. Are they all needed?

Response: Thank you for your comments. we have to admit that the supplementary figures did not bring new information to the manuscript, while it is the raw data for our manuscript. After our careful discussion, we decided to keep them in our revised manuscript. Keeping them could allow our findings easily to be repeated by some readers. Thanks a lot.

-What sort of vaccines are BNT162b2 and BBIBP-CorV. This isn’t provided. You also do not define “mix”

Response: Thanks for your excellent suggestions. As you suggested, we have added the definition in our revised methods. Below is the revision: “Inactivated vaccines include BBIBP-CorV, Corona Vac, or Sinopharm; mRNA vaccines consist of BNT162b2 or mRNA-1273; adenovirus vaccines are comprised of ChA-dOx1 nCoV-19 or Ad.26.COV2.S; and mixed vaccines mean more than one type of vaccine.” Thanks a lot.

-The initial inclusion of “Wagner A, Garner-Spitzer E, Schötta AM, et al” was clearly your error. It should be omitted.

Response: Thanks for your suggestions. As you suggested, we have made the corrections in our revised manuscript, Figure 1, and Table S3. Thanks a lot.

-Table S4 lacks an informative legend

Response: Thanks for your comment. We have revised the legends of Table S4 as bellow: “Meta-regression for odds ratio for the seroconversion in COVID-19 vaccinated PLWH with high and low CD4 T cells.”

Round 2

Reviewer 3 Report

Take care to write for an audience who are not familiar with the software used but do care about the conclusions

Author Response

- Take care to write for an audience who are not familiar with the software used but do care about the conclusions

Response: Thanks for your reminder. As we responded previously, the results were consistently using RevMan or R software. We also provided the raw data in the supplementary and the conclusions were also easily replicated. Actually, R software is a common software to analyze the data in the meta-analysis due to the excellent visualization effects. We also appreciate your comments. Thanks a lot.